# Interpretable machine learning for automated left ventricular scar quantification in hypertrophic cardiomyopathy patients

Zeinab Navidi[1,2,3☯], Jesse Sun[1☯], Raymond H. Chan[1], Kate Hanneman[4], Amna Al-Arnawoot[4], Alif Munim[3], Harry Rakowski[1], Martin S. Maron[5], Anna Woo[1], Bo Wang[1,2,3,6‡], Wendy Tsang[1‡]*

1 Division of Cardiology, Peter Munk Cardiac Center, Toronto General Hospital, University Health Network, University of Toronto, Toronto, Canada, 2 Department of Computer Science, University of Toronto, Toronto, Canada, 3 Vector Institute, Toronto, Canada, 4 Department of Radiology, University Health Network, University of Toronto, Toronto, Canada, 5 Division of Cardiology, Tufts Medical Center, Boston, United States of America, 6 Department of Laboratory Medicine and Pathobiology, University of Toronto, Toronto, Canada

☯ These authors contributed equally to this work.
‡ BW and WT also contributed equally to this work.
* wendy.tsang@uhn.ca

**Data Availability Statement:** Data cannot be shared publicly because of HIPAA requirements. For data requests, please contact the

## Abstract

Scar quantification on cardiovascular magnetic resonance (CMR) late gadolinium enhancement (LGE) images is important in risk stratifying patients with hypertrophic cardiomyopathy (HCM) due to the importance of scar burden in predicting clinical outcomes. We aimed to develop a machine learning (ML) model that contours left ventricular (LV) endo- and epicardial borders and quantifies CMR LGE images from HCM patients.We retrospectively studied 2557 unprocessed images from 307 HCM patients followed at the University Health Network (Canada) and Tufts Medical Center (USA). LGE images were manually segmented by two experts using two different software packages. Using 6SD LGE intensity cutoff as the gold standard, a 2-dimensional convolutional neural network (CNN) was trained on 80% and tested on the remaining 20% of the data. Model performance was evaluated using the Dice Similarity Coefficient (DSC), Bland-Altman, and Pearson's correlation. The 6SD model DSC scores were good to excellent at 0.91 ± 0.04, 0.83 ± 0.03, and 0.64 ± 0.09 for the LV endocardium, epicardium, and scar segmentation, respectively. The bias and limits of agreement for the percentage of LGE to LV mass were low (-0.53 ± 2.71%), and correlation high (r = 0.92). This fully automated interpretable ML algorithm allows rapid and accurate scar quantification from CMR LGE images. This program does not require manual image pre-processing, and was trained with multiple experts and software, increasing its generalizability.

## Author summary

Accurate scar quantification of cardiac magnetic resonance (CMR) late gadolinium enhancement (LGE) images is important in managing hypertrophic cardiomyopathy

corresponding author at wendy.tsang@uhn.ca. All code is available in a public link: https://drive.google.com/drive/folders/1197aHAFmLWqknuvrKAm51i7fH4q8c7Bo and will be shared on GitHub after publication.

**Funding:** Funding for this study was provided by the Peter Munk Cardiology Center Innovation Fund and the MSH-UHN AMO Innovation Fund. BW is partially supported by the CIFAR AI Chair Program. WT is supported by a Heart and Stroke Foundation of Canada National New Investigator Award. The funders had no role in study design, data collection and analysis, decision to publish, or preparation of the manuscript.

**Competing interests:** The authors declare that they have no competing interests.

(HCM) patients. We developed a 2D convolutional neural network to quantify CMR LGE in HCM patients that is computationally interpretable and trained using multicenter data analyzed by 2 expert readers using 2 different analysis packages. Our model demonstrated low bias and limits of agreement and high correlation with expert analysis. Benchmarking comparison was performed between our algorithm and standard U-Net model with and without cropped raw images. Our method showed superior performance and has high potential for clinical adaptability.

## Introduction

Hypertrophic cardiomyopathy (HCM) is the most common inheritable cardiomyopathy with a reported prevalence as high as 1 in 200 [1]. Patients with HCM can develop myocardial fibrosis, which is associated with heart failure and sudden cardiac death [2–5]. Late gadolinium enhancement (LGE) techniques on cardiovascular magnetic resonance (CMR) imaging allow for non-invasive detection and quantification of fibrosis in patients with HCM. Due to its prognostic value, current guidelines for the management of HCM patients recommends assessment of LGE by CMR as an important component for risk stratification [6–8]. However, in current practice, LGE quantification can be subjective, time-consuming, and requires training to delineate both the myocardial borders and the hyper-enhanced regions on the LGE images [9–12]. These issues are even more pronounced in HCM patients where scar is most often patchy and multi-focal [12].

Recently, machine learning (ML) and specifically deep convolutional neural networks (CNN) have been used to automate CMR LGE image segmentation in HCM patients [13–18]. However, many of these ML algorithms require image pre-processing or relied on a single expert reader as their reference standard, potentially limiting their generalizability and adoptability. The Shape Attentive U-Net (SAUNet) model, a previously developed algorithm from our group, focuses on model interpretability and robustness, and has shown promising performance on medical image segmentation [19]. We aimed to use SAUNet to develop a 2-dimensional (2D) computationally interpretable CNN model to efficiently and accurately segment left ventricular (LV) endo- and epicardial borders and quantify scar on LGE CMR images in HCM patients with minimal pre-processing and using a single NVIDIA Tesla P100 graphics card.

## Results

Baseline patient demographics are presented in Table 1. The median age of patients was 52 years (interquartile range, IQR, 39–61 years) and the majority (70%) were male. The short axis LGE images consisted of 8 ± 2 images per patient (interquartile range, IQR, 7–9 images). LGE images from 307 HCM patients were divided into exclusive training or testing subsets. The training set included 247 patients (2056 images) with 200 patients having LGE scar (927 images). The testing set consisted of 60 patients (501 images) with 53 patients having LGE scar (253 images). See the S1 Table for more detailed information.

### Model development

Model development is described in Fig 1. The average analysis time for one image using our algorithm was less than 70 milliseconds using a single NVIDIA Tesla P100 GPU. Figs 2 and 3 provide examples of the expert-based analysis and contours predicted by the

**Table 1. Patient demographics.**

| Characteristic | Toronto (n = 211) | Boston (n = 96) | All (n = 307) |
|---|---|---|---|
| Age (y)* | 52 (41, 61) | 49 (35, 59) | 52 (39, 61) |
| Male sex | 152 (72%) | 64 (66%) | 216 (70%) |
| Body surface area ($m^2$)* | 1.96 (1.81, 2.09) | 1.93 (1.75, 2.13) | 1.96 (1.8, 2.11) |
| LV mass indexed to BSA ($Kg/m^2$)* | 0.06 (0.05, 0.08) | 0.07 (0.06, 0.08) | 0.06 (0.05, 0.08) |
| Maximum LV wall thickness (mm)* | 18.3 (15.25, 22) | 18.8 (16.3, 22.23) | 18.3 (15.72, 22.1) |
| New York Heart Association classification | | | |
| 1 | 111 | 43 | 154 |
| 2 | 76 | 31 | 107 |
| 3 | 22 | 22 | 44 |
| 4 | 1 | 0 | 1 |
| Coronary artery disease | 20 | 8 | 28 |
| Atrial fibrillation | 22 | 16 | 38 |
| Risk factors for sudden death | | | |
| Non-sustained ventricular Tachycardia | 34 | 11 | 45 |
| Unexplained syncope | 24 | 8 | 32 |
| Family history of sudden cardiac death | 35 | 12 | 47 |
| Left ventricular outflow tract obstruction | 42 | 39 | 81 |

* sign means the variable is represented in median (first quartile, third quartile) format. Other numbers in the table represent the number of patients (BSA = Body Surface Area, LV = Left Ventricular).

SAUNet algorithm in patients with and without scar. The heatmaps obtained from different layers of the model were obtained to visualize the focus of the model at different steps. These intermediate-level outputs for each layer of the SAUNet model aided in identifying which layers required modification to improve the final segmentation performance for faulty predictions. The spatial attention maps of the final Dual Attention Block (DA-Block) are provided in Figs 2 and 3 for the corresponding samples to highlight regions of interest by the model during intermediate stages of the algorithm computation [19].

## Model LV Segmentation and Scar Quantification

**4SD SAUNet model.** LV segmentation by the developed model demonstrated excellent similarity to manual segmentation with a DSC score of 0.92 ± 0.04 for the LV endocardium and 0.83 ± 0.03 for the LV epicardium. For LV LGE scar quantification, the DSC score was good at 0.60 ± 0.08. There was no significant difference between the average LGE scar mass quantified by the 4SD model compared to manual expert analysis (3.77 ± 7.11 g model versus 4.56 ± 7.23 g expert, p = 0.55, Table 2). The per patient correlation between SAUNet 4SD model LGE scar mass and expert manually quantified LGE scar mass was high (r-value 0.92, Fig 4A). Bland-Altman analysis for scar mass demonstrated that the 4SD model had a low bias of -0.79 g with limits of agreement of -6.26 g to 4.68 g (Fig 4B).

The percentage LGE quantified by the SAUNet 4SD model was 2.55 ± 4.94% compared to 3.47 ± 5.62% obtained by manual expert-analysis, which was not significantly different (p = 0.34). Correlation between SAUNet 4SD model and manually quantified percentage LGE scar was high (r-value 0.90, Fig 4C). For %LGE, Bland-Altman analysis demonstrated that the 4SD model slightly underestimated LGE with a bias of -0.93% and limits of agreement of -5.64% to 3.78% (Fig 4D). There was no significant difference between sites in either scar quantification or the percentage of LGE quantified.

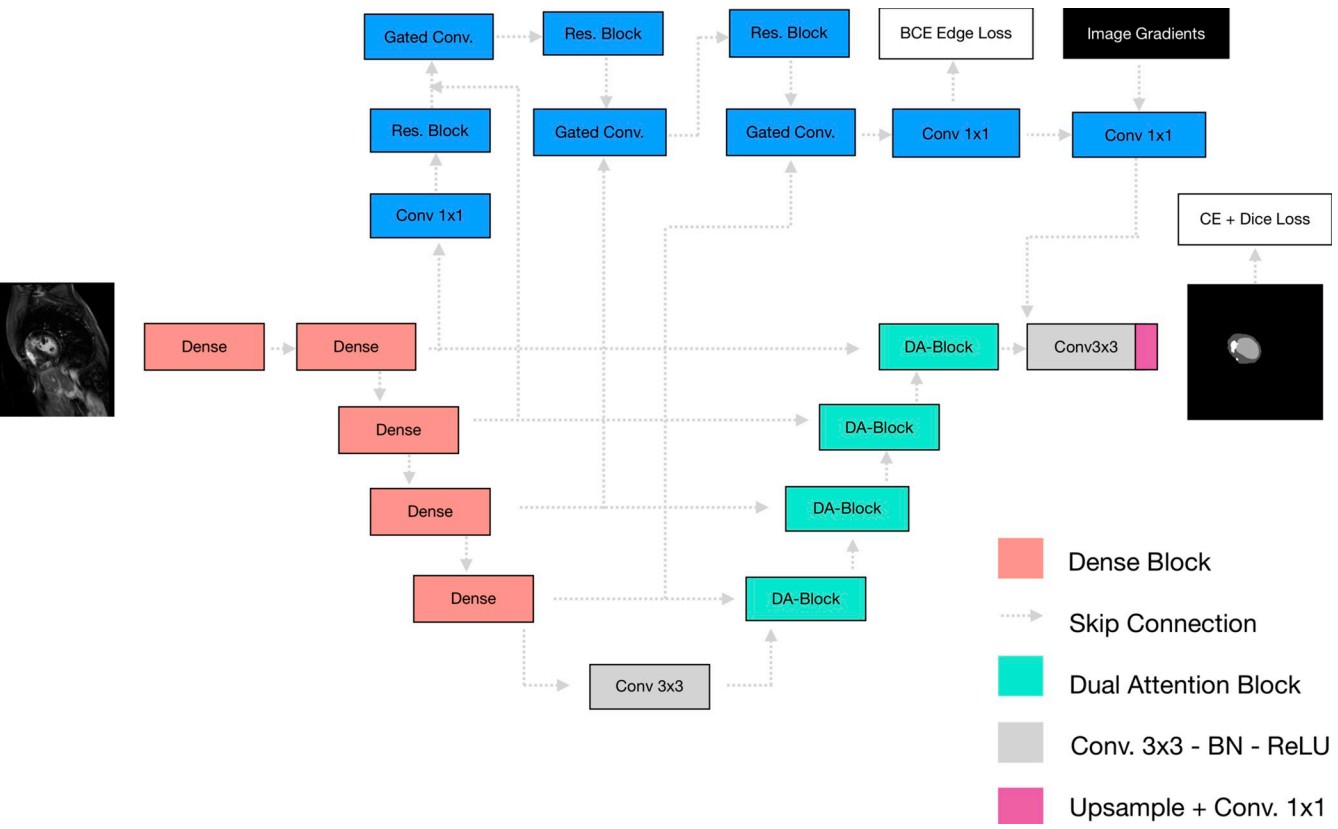

**Fig 1.** Diagram of the 2D CNN model (SAUNet) used in this study with sample input and output (BCE = Binary Cross Entropy, CE = Cross Entropy, Conv = Convolutional Block, DA-Block = Dual Attention Block, Res. Block = Residual Block).

**6SD SAUNet model.** LV segmentation by the SAUNet 6SD model demonstrated excellent similarity to manual segmentation with a DSC score of 0.91 ± 0.04 for the LV endocardium and 0.83 ± 0.03 for the LV epicardium. For LV LGE scar quantification, the DSC score was good at 0.64 ± 0.09. There was no significant difference between the average LGE scar mass quantified by the 6SD model compared to manual expert analysis (2.21 ± 4.38 g model versus 2.68 ± 4.77 g, p = 0.50, Table 2). The per patient correlation between the 6SD model LGE scar mass and expert manually quantified LGE scar mass was high with an r-value of 0.91 (Fig 5A). Bland-Altman analysis for scar mass demonstrated that the 6SD model had a low bias of -0.56 g with limits of agreement of -4.44 g to 3.32 g (Fig 5B).

The percentage LGE quantified by the SAUNet 6SD model was 1.28 ± 2.76% compared to 1.80 ± 3.38% obtained by manual expert-analysis, which was not significantly different (p = 0.35). Correlation between the 6SD model and manually quantified percentage LGE scar was high with an r-value of 0.92 (Fig 5C). Bland-Altman analysis for LGE% demonstrated that the 6SD model had a low bias of -0.53 g with limits of agreement of -3.23 g to 2.18 g (Fig 5D).

**U-Net model comparison.** Tables 3–4 provides the comparisons for LV segmentation and scar detection between the 4SD and 6SD SAUNet models and the U-Net models. While the DSC scores for SAUNet and U-Net endo and epicardium segmentation were close, the DSC scores for scar prediction were 5–7% higher for 4SD and 6SD SAUNet models compared to the implemented U-Net models. Cropping the images to isolate the LV region improved the absolute DSC scores for both the SAUNet and U-Net models by 2–8% compared to their

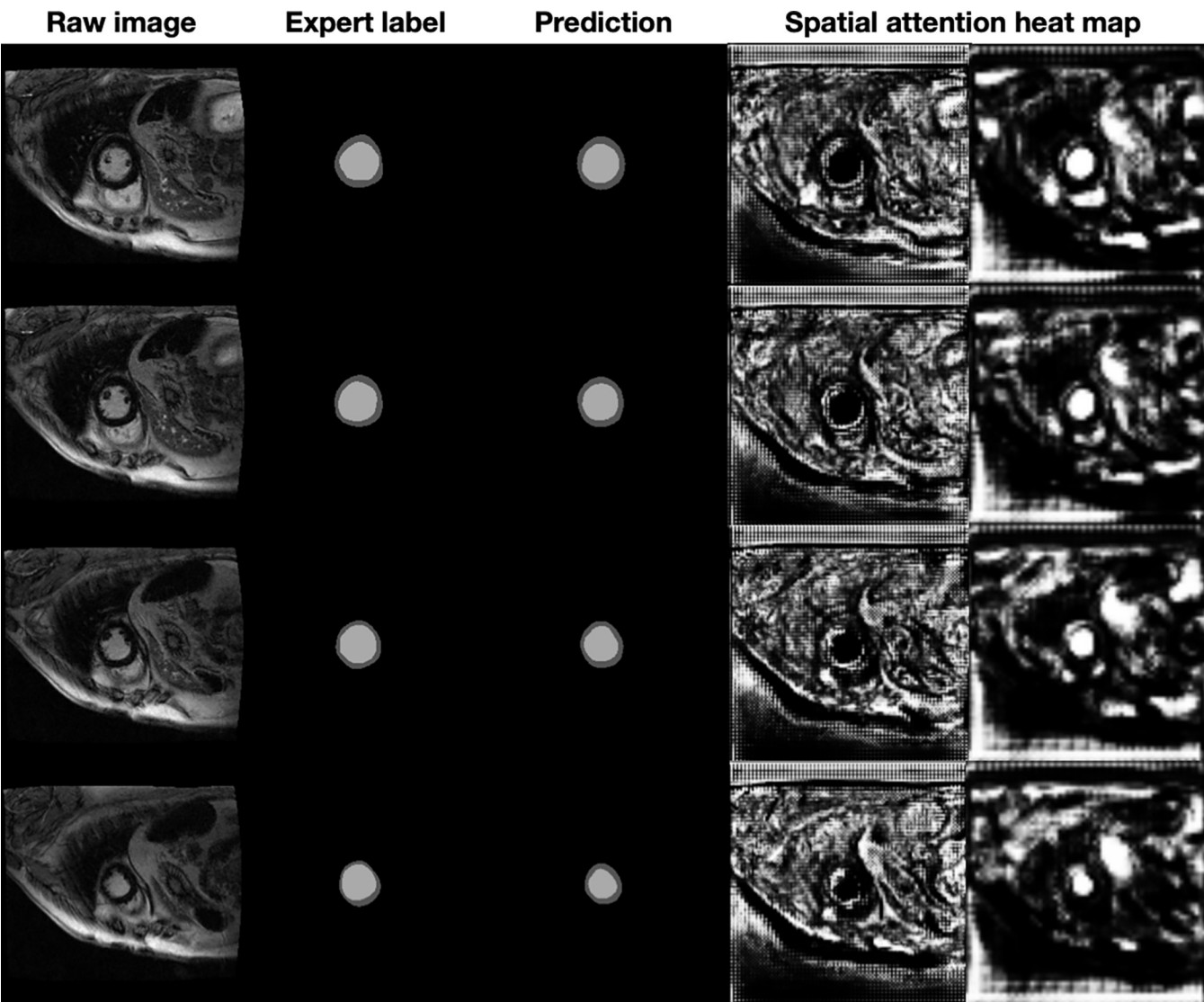

**Fig 2.** Examples of patients with no scar. The leftmost column is the original MRI image, the second column is the expert-based label or ground truth, and the third column is the model prediction. The fourth and fifth columns are the spatial attention heatmaps at the 1/2 and 1/4 resolutions, respectively.

respective results without image cropping. However, the results of the SAUNet program remained superior to U-Net (Table 3).

Bland Altman analysis, t-test, predicted mean mass, and expert-read mean mass and LGE% comparing SAUNet against U-Net for the 4SD and 6SD models are provided in Table 4 (see S1 Fig and S2 Fig for more analyses).

**Sex comparison.** The mass and LGE% statistics of ground truth from different sex groups are provided in S2 Table, using the 6SD SAUNet model. By Welch's t-test, there are no differences between the predicted and ground truth means for between sexes.

## Discussion

In this study, we have used multicenter CMR data to successfully develop and validate a fully automated deep learning algorithm that contours the LV endo- and epicardial borders and quantifies LGE in patients with HCM. Based on the experiments we performed, our pipeline

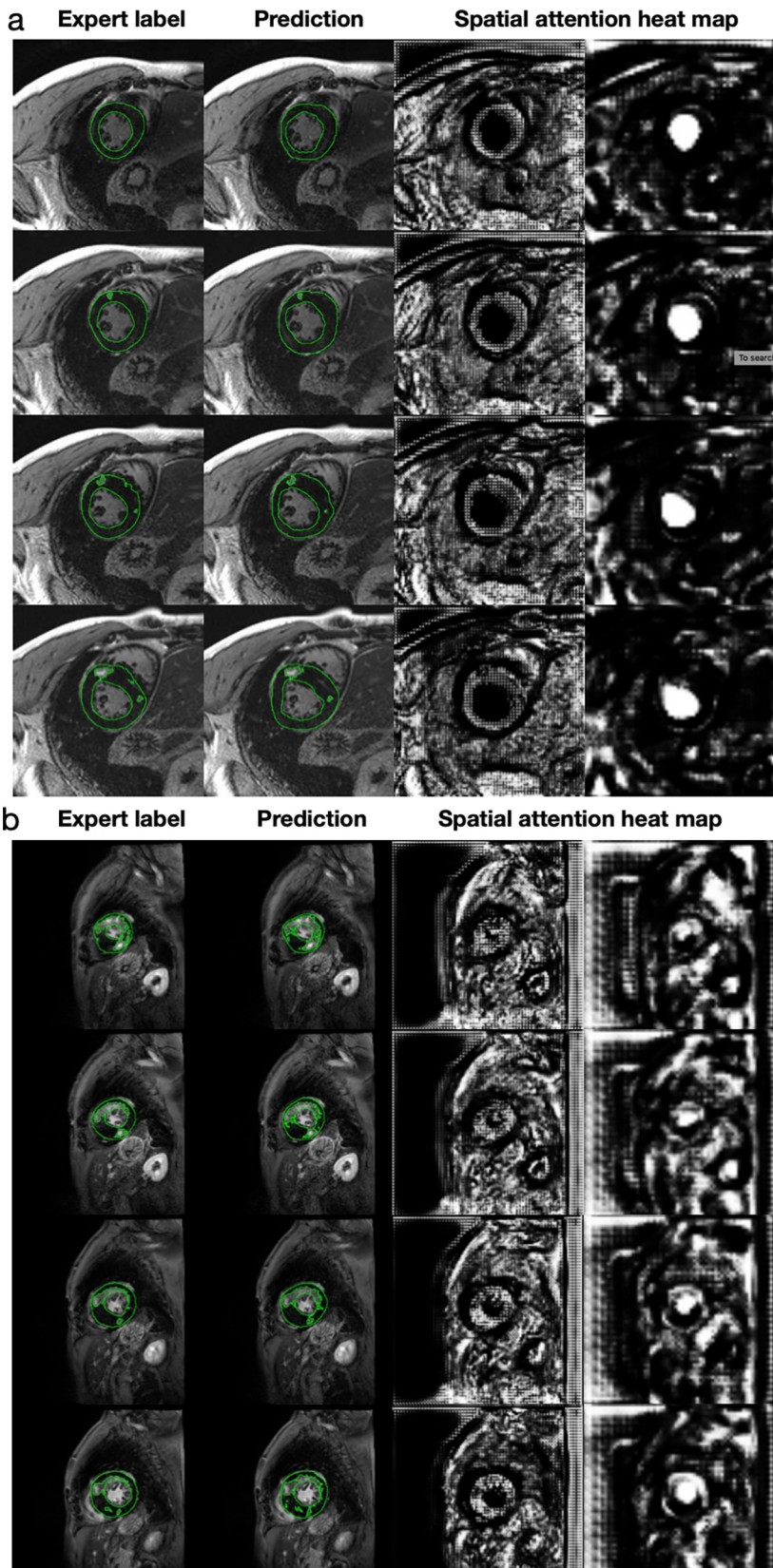

**Fig 3.** Examples of patients with mild (8% LGE) (a) and large (51% LGE) scar burden (b). The left most column is the expert-based label or ground truth, and the second column is the model prediction. The third and fourth columns are the spatial attention heatmaps at the 1/2 and 1/4 resolutions, respectively. Note, not all slices from each patient are presented.

provides more accurate and robust scar quantification as it was trained with data from different sites, vendors, readers, and analysis packages. It also has higher clinical utility as it does not require manual image pre-processing and can rapidly analyze standard CMR LGE images using a single graphics card. Finally, it is based on SAUNet architecture, allowing strong computational interpretability by providing visualization attention maps during the intermediate stages of the segmentation.

Compared to previous studies specifically investigating HCM patients, the CMR images used in our study were collated from multiple vendors, and two distinct sets of these images were analyzed by two different readers using different software packages. The incorporation of data from different sites, vendors, readers, and analysis packages enabled us to develop a more robust model. Studies have demonstrated significant inter-reader variability in CMR LGE image analysis, which is exacerbated by the patchy multifocal CMR LGE appearance in HCM patients [9–12]. As such, incorporating more than one reader reduces the risk of bias that may develop in a deep learning model trained using contours from one clinician. Given the aforementioned advantages, the dataset used to train and test the algorithm allowed us to develop a model with greater potential for wider clinical use.

Moreover, we trained and validated our model to function efficiently on a single NVIDIA GPU to analyze uncropped images. Our algorithm can rapidly process one image in less than 0.07 seconds, which is comparable to previous programs. Using our program, an average CMR study consisting of approximately 8 LGE images would require less than 0.56 seconds to be analyzed. This is considerably shorter than the time currently required for experts to manually segment the LV and quantify the scar burden from CMR LGE images [20]. This offers time-savings to clinicians and will reduce the amount of training required to perform CMR LGE scar analysis. It will also increase the number of patients receiving quantitative scar burden measurement over a qualitative assessment.

Our pipeline does not require the extra step of manually cropping the images such that all the structures around the LV are removed. Most of the previously developed programs were trained and tested using these LV focused CMR LGE images, because it reduces computational requirements as the tasks (i.e., identify the LV and then segment the LV) required by the algorithm are reduced. The ability to analyze a more complete image permits this program to be more easily integrated into an automated clinical workflow.

To demonstrate the improvements gained by our SAUNet-based algorithm, we compared our results against a U-Net model trained and tested on the same samples with the same gold standard definition. Our benchmarking also included assessing the impact of image cropping

**Table 2. Expert-based CMR LGE quantification versus CNN model LGE prediction (4SD and 6SD SAUNet models).**

| | Expert CMR LGE Quantification (Mean ± SD) | CNN Model Prediction (Mean ± SD) | Correlation | Bias | LOA (2SDs) |
|---|---|---|---|---|---|
| 4SD model | | | | | |
| LGE mass | 4.56 ± 7.23 g | 3.77 ± 7.11 g | 0.92 | -0.79 g | 5.47 g |
| %LGE volume | 3.47 ± 5.62% | 2.55 ± 4.94% | 0.90 | -0.93% | 4.71% |
| 6SD model | | | | | |
| LGE mass | 2.68 ± 4.77 g | 2.21 ± 4.38 g | 0.91 | -0.56 g | 3.88 g |
| %LGE volume | 1.80 ± 3.38% | 1.28 ± 2.76% | 0.92 | -0.53% | 2.71% |

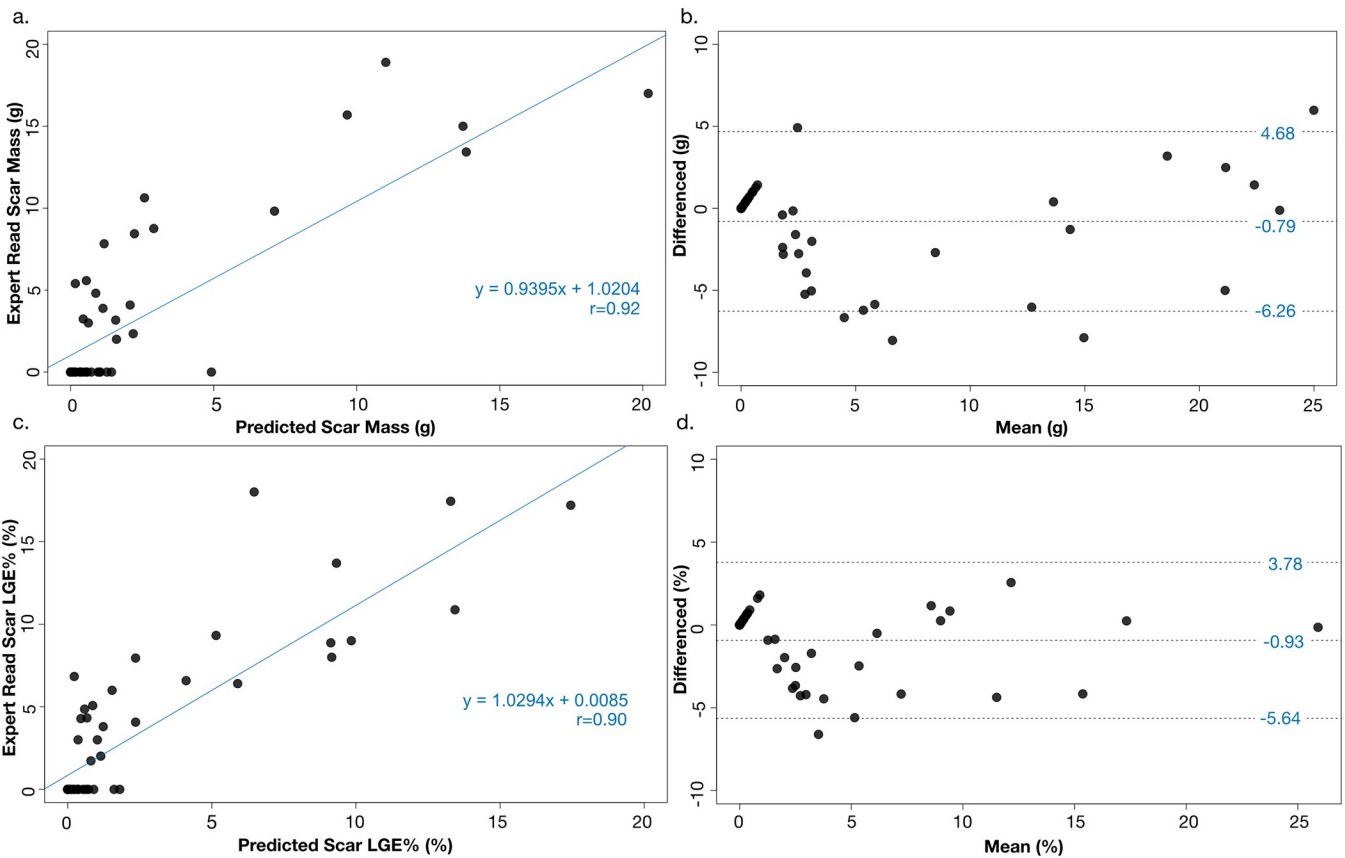

**Fig 4.** Correlation (a) and Bland-Altman analysis (b) between the expert-based manual analysis and the model prediction for CMR LGE scar mass. Correlation (c) and Bland-Altman analysis (d) between the expert-based manual analysis and model prediction for percentage of LGE volume.

to isolate the LV. Overall, CMR LGE quantification is a more complicated task as demonstrated by the lower DSC scores compared to LV segmentation as seen in this analysis and in prior publications. However, our SAUNet model performed better at scar segmentation compared to the standard U-Net, and this improvement persisted with cropped images. Overall, SAUNet results in an algorithm that outperforms standard U-Net architecture [16].

The use of SAUNet also improves both computational interpretability and performance of final prediction. SAUNet is a 2D architecture that we developed that uses shape-dependent information in addition to texture information to improve a CNN model's robustness [19]. We have taken this architecture and adapted it to HCM LV segmentation and LGE quantification. Previously developed HCM algorithms used off-the-shelf U-Net models, which lack interpretability. In comparison, SAUNet allows for multi-level computational interpretability and removes the need for additional post-hoc computations or gradient-based saliency methods for sensitivity analysis [19,21]. The intrinsic attention maps within the SAUNet model are a strong alternative to saliency methods with some improved computational advantages [19]. Namely, attention maps at varying resolutions and layers of the model are all computed during the forward-pass of an image, reducing the need for multiple iterations of additional post-hoc computation to compute the saliency map of different points within the model. We suggest this computational interpretability is beneficial as having a seamless method to visualize comprehensible intermediate stages of the model is useful for debugging the pipeline, especially in the case of using larger datasets with greater variance (i.e., from different readers and centres).

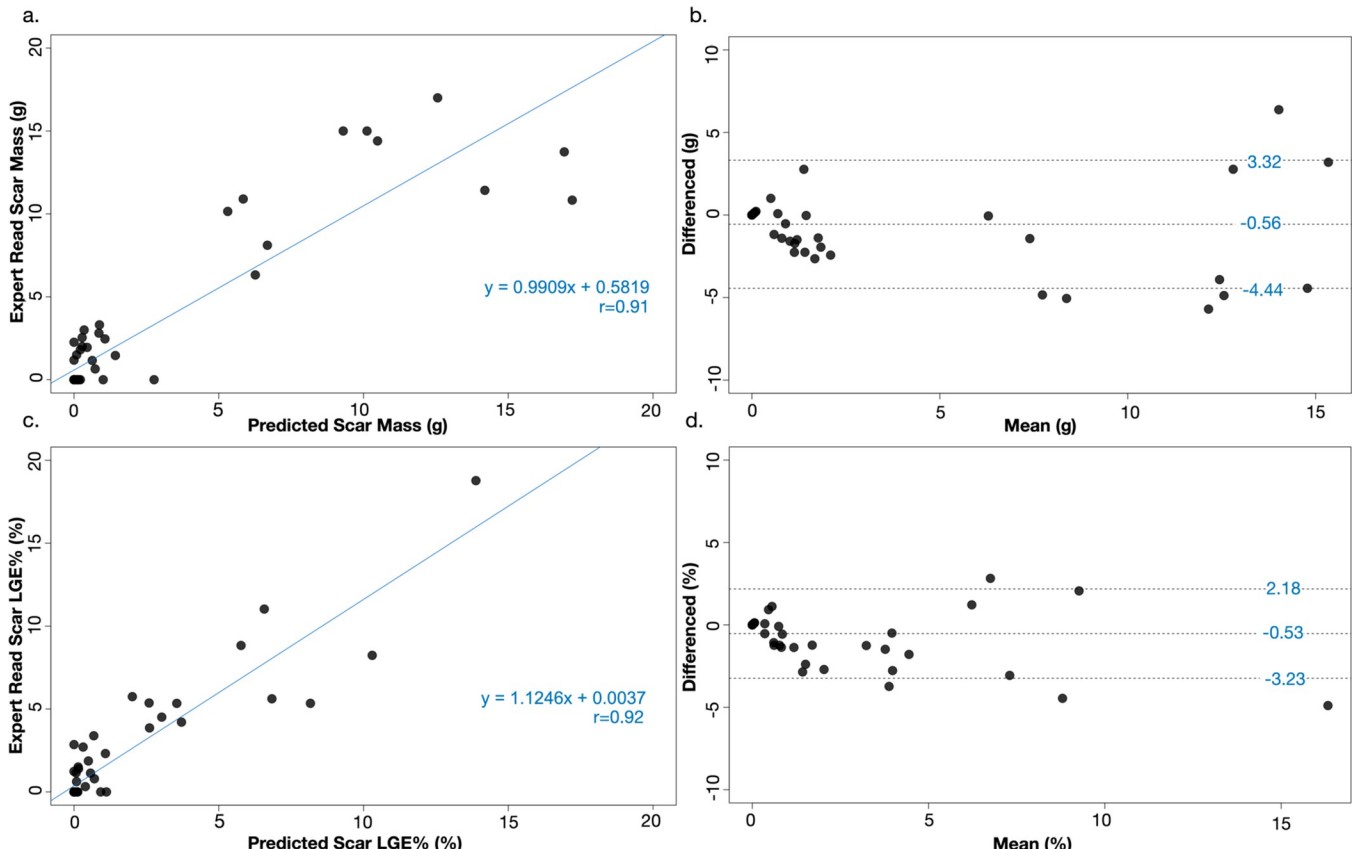

**Fig 5.** Correlation (a) and Bland-Altman analysis (b) between the expert-based manual analysis and the model prediction for CMR LGE scar mass. Correlation (c) and Bland-Altman analysis (d) between the expert-based manual analysis and model prediction for percentage of LGE volume.

Existing works for automated scar quantification in HCM patients proposed programs with no framework or guidelines for interpretability analysis [13–16]. By verifying that an algorithm is not perpetuating biases, a valuable tool can be created to help solve the challenges numerous clinicians face in medical image analysis.

## Limitations

One limitation to this study is the slight pixel intensity shifts seen within a patient's CMR LGE images. Often expert readers will define a region of interest of normal myocardium on a

**Table 3. Performance comparison between SAUNet and U-Net models on endo- and epicardial and scarring segmentation on HCM patients.**

| Dice Score Coefficient | | Endocardium | Epicardium | Scar |
|---|---|---|---|---|
| SAUNet | 4SD | **0.92 ± 0.04** | 0.83 ± 0.03 | 0.60 ± 0.08 |
| | 6SD | 0.91 ± 0.04 | 0.83 ± 0.03 | 0.64 ± 0.09 |
| U-Net | 4SD | 0.91 ± 0.04 | 0.81 ± 0.03 | 0.55 ± 0.07 |
| | 6SD | 0.91 ± 0.04 | 0.80 ± 0.03 | 0.57 ± 0.08 |
| SAUNet cropped version | 4SD | 0.91 ± 0.02 | 0.83 ± 0.02 | 0.62 ± 0.07 |
| | 6SD | 0.91 ± 0.02 | **0.84 ± 0.02** | **0.68 ± 0.09** |
| U-Net cropped version | 4SD | 0.90 ± 0.02 | 0.82 ± 0.02 | 0.61 ± 0.07 |
| | 6SD | 0.90 ± 0.02 | 0.82 ± 0.02 | 0.65 ± 0.08 |

**Table 4. Bland Altman analysis, t-test, predicted mean mass, and expert-read mean mass and LGE% comparing SAUNet against U-Net for the 4SD and 6SD models.**

| | Bias + LOA | T-test P-value | Predicted mean + SD | Expert-read mean + SD |
|---|---|---|---|---|
| **Mass** | | | | |
| 4SD SAUNet | -0.79 ± 5.47 g | 0.55 | 3.77 ± 7.11 g | 4.56 ± 7.23 g |
| 4SD U-Net | -0.72 ± 7.08 g | 0.58 | 3.84 ± 6.91 g | |
| 6SD SAUNet | -0.56 ± 3.88 g | 0.50 | 2.21 ± 4.38 g | 2.68 ± 4.77 g |
| 6SD U-Net | -0.91 ± 4.68 g | 0.35 | 1.77 ± 3.58 g | |
| **LGE%** | | | | |
| 4SD SAUNet | -0.93 ± 4.71% | 0.34 | 2.55 ± 4.94% | 3.47 ± 5.62% |
| 4SD U-Net | -1.24 ± 5.95% | 0.17 | 2.23 ± 3.47% | |
| 6SD SAUNet | -0.53 ± 2.71% | 0.35 | 1.28 ± 2.76% | 1.80 ± 3.38% |
| 6SD U-Net | -0.72 ± 3.45% | 0.18 | 1.08 ± 2.40% | |

limited number of slices (i.e., not on every LGE image). Thus, subtle intensity shifts to the overall pixel values on slices where the normal region was not defined could lead to over- or underestimation of scarring in the ground truth mask. While the extracted pixels' statistics generally performed well, providing a normal myocardial region contour for every LGE slice would enhance the accuracy of individual slices' labelling. Validation with more interpreters or with pathologic confirmation would be important in removing potential bias and human error in the ground truth contouring. Finally, external validation of this program would be valuable to assess wider generalizability.

## Conclusions

Using novel machine learning methods, we have successfully developed an automatic deep learning pipeline that rapidly provides LV endo- and epicardial segmentation and LV scar quantification on CMR images. Our program was developed with multiple experts and software packages. It also does not require heavy image pre-processing and has greater interpretability, potentially allowing it to be integrated into routine clinical practice.

## Materials and methods

### Study patients

The study design is summarized in Fig 6. We retrospectively studied 2557 CMR LGE images from 307 HCM patients imaged at the University Health Network (Toronto, Canada) and Tufts Medical Center (Boston, USA) between November 2001 and September 2015.

The diagnosis of HCM was made clinically as per societal guidelines [7,8,22]. Baseline demographics and clinical characteristics were obtained from the patient's electronic medical record. Echocardiograms were reviewed for information regarding the presence of LV outflow tract obstruction. This study received the proper ethical oversight and was approved by the Research Ethics boards of the University Health Network (Toronto, Canada) and Tufts Medical Center (Boston, USA).

### CMR imaging

CMR imaging was performed using 1.5T or 3T scanners (Achieva, Philips, the Netherlands; or Avanto/Skyra_fit/Signa Excite/Verio, Siemens, Germany) using steady-state, free-precession breath-hold cines in sequential short-axis slices from the atrioventricular ring to the apex (6–8 mm slices with 0–2 mm inter-slice gap). LGE images were acquired 10 to 20 minutes after

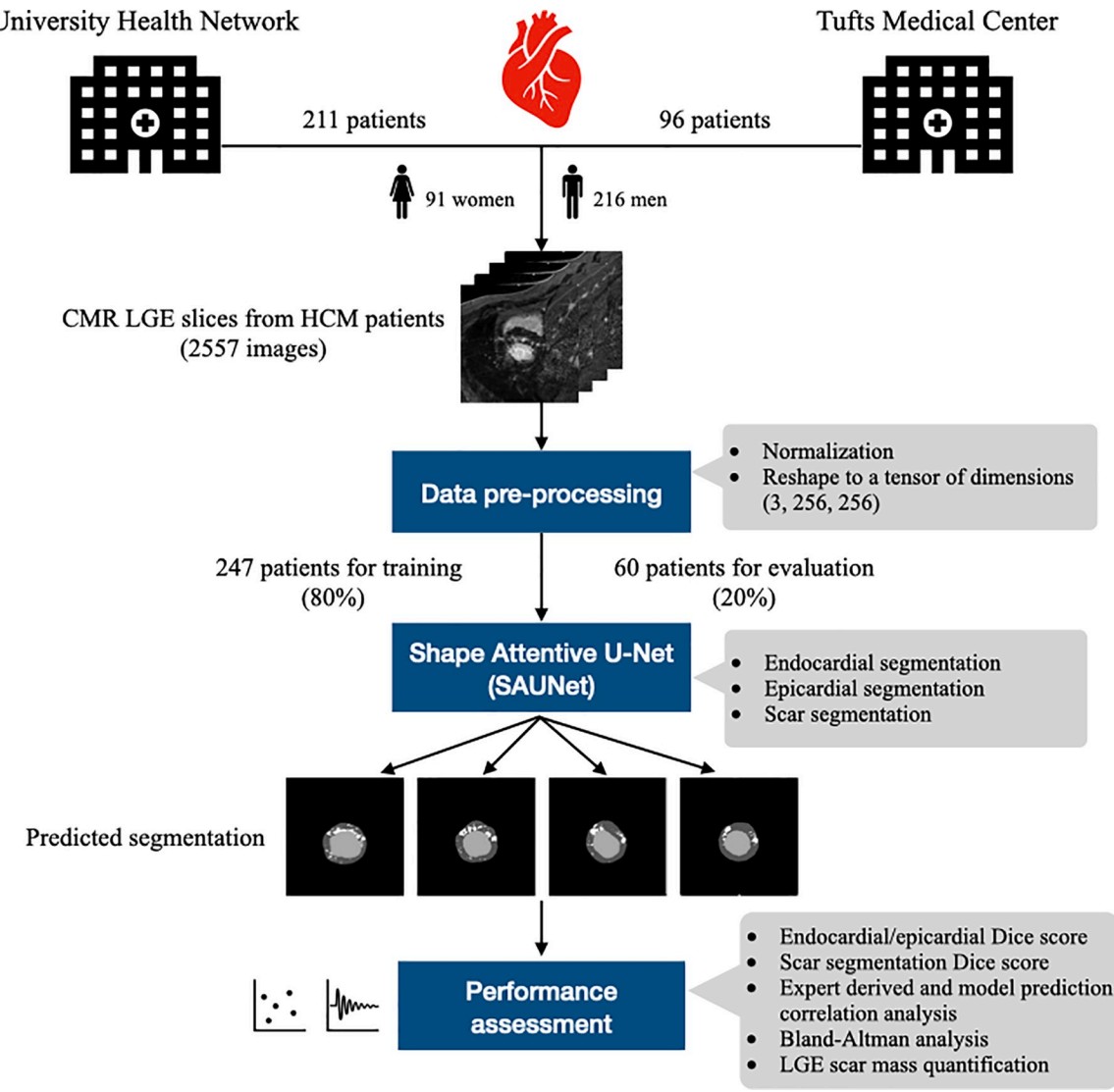

**Fig 6. Central illustration of data processing, modelling, and assessments of this study.**

intravenous administration of 0.15–0.2 mmol/kg gadolinium-DTPA with breath-hold 2D phase-sensitive inversion-recovery sequences in identical planes as in the cine images. Inversion time was optimized to null normal myocardial signal. For phase-sensitive sequences, uncorrected magnitude images were used.

## CMR image analysis

LV endo- and epicardial contouring and LGE scar quantification were performed using commercially available software packages (QMASS DSI version 7.4, Medis Medical Imaging, Leiden, Netherlands; CVi42 5.11.1, Circle Cardiovascular Imaging, Calgary, Canada) (S3 Table). LV endocardial and epicardial borders were manually contoured on the LGE images to define the myocardium, taking care to exclude papillary muscles and the trabecular blood pool. LGE quantification was performed by 2 expert readers (R.H.C., A.A) [4,23,24]. Briefly, LV myocardial areas with pixel intensity values 4 standard deviations (SD) greater than the mean of a

normal myocardial region that was manually defined by experts, were considered scar, and labelled as the 4SD LGE ground truth. A 6SD ground truth was also generated using a 6SD cut-off. The LGE area of each slice was then summed across each patient's MRI images to generate the total LGE mass. Inter- and intra-observer agreement using this method has previously been published [4].

For ML, each patient's CMR study contained manually traced contours corresponding to the LV endo- and epicardium that was used for LV segmentation, and a contour identifying a region of normal myocardium on the LGE images. The mean and standard deviations of the normal myocardial region were used to calculate and label the scarred region for that patient's LGE images. Each 2D image was converted to a tensor and then normalized across the slice before it was fed into the model during training and validation. Due to the downsizing and upsizing nature of the architecture of the deep model used in this study, each image was reshaped to the size of 256 by 256 pixels prior to analysis. To avoid training bias, the datasets for each site were randomly split into training and testing data sets in a 4:1 ratio, ensuring that the 80% of cases with LGE present were included in the training set and the remaining 20% were used for validation. A model was trained using the 4SD ground truth and separately the 6SD ground truth label.

## Convolutional neural network

An interpretable version of the 2D U-Net deep neural network (DNN) model called Shape Attentive U-Net (SAUNet) was used in this study. SAUNet architecture is composed of two main streams, including the shape stream that processes boundary information and the texture stream. The shape stream is composed of gated convolutional layers and residual layers. The gated convolutional layers are used to fuse texture and shape information, while the residual layers are used to fine-tune the shape features [19].

Each image sample was processed as a (3, 256, 256) tensor, which was fed into the network. The SAUNet model deploys an encoder-decoder based architecture to learn the features of LV and scar segmentation, and outputs a (4, 256, 256) tensor. Each of the four channels in the output tensor corresponds to one of the 4 segmentation classes: background, LV epicardium, LV endocardium, and scar. We apply a Softmax function across the channel dimension such that each tensor value in a given channel corresponds to the probability that corresponding pixel is labelled its respective channel's class. Thus, our model's final prediction is computed as the argmax of the output tensor after the SoftMax function. In other words, each pixel is labelled as the class that has the highest probability across the channels for the respective pixel.

We deployed transfer learning and data augmentation techniques to train a robust and generalizable model and speed up the learning process. We started with a SAUNet model, whose initial parameters were determined from training to segment the left and right ventricular cavities and left myocardium on CMR data on the AC17 dataset [19,25]. Multiple real-time augmentation methods were deployed during the training step to prevent overfitting and increase the generalizability of the model. By applying various data augmentation techniques, multiple potential variations of the provided images were seen by the model, helping mitigate the data hunger of deep models [26,27]. Random cropping, flipping, rotating, and random elastic deformation filtration were applied on the images to allow the deep model to identify various shapes of images [28].

Mini-batch gradient descent was deployed during training, which provided fast training convergence while limiting memory usage. After performing hyperparameter search, we used a batch size of 4 for all model training. SAUNet includes a combination of three loss functions as the total loss composed of the segmentation loss and the shape stream boundary loss. SAUNet uses a Cross-Entropy loss as a commonly used loss function for image segmentation tasks

($L_{CE}$), Dice loss which measures the overlap and similarity between two sets for each class ($L_{Dice}$), and an extra loss called Edge loss which denotes the binary cross entropy loss of the predicted shape boundaries ($L_{Edge}$). The total loss, $L_{Total}$, is defined as,

$$L_{Total} = L_{CE} + L_{Dice} + L_{Edge}.$$

The stochastic gradient descent (SGD) algorithm was deployed for training optimization with a momentum of 0.9 and a learning rate value initialized to 0.0005. For regularization, a weight decay of 1e-4 was used, and the best model parameters were selected based on the scarring segmentation Dice Similarity Coefficient (DSC) score between 180 epochs of training.

Image processing and performance evaluation were performed using Python (version 3.6; Python Software Foundation) in the PyTorch framework. All analysis was performed using a single NVIDIA Tesla P100 graphics card. Model implementation and their parameters are available online on our GitHub page.

### Scar quantification

On the CMR LGE images, we computed the scar and no-scar myocardial mass (in grams) by multiplying the voxel volume by the number of pixels labelled scar or no-scar, respectively. Summation of segmented no-scar and scarred myocardial mass was used to calculate the total LV mass. The ratio of scar mass to total LV myocardium mass (%LGE) was computed for automatic and manual segmentations. Scar regions bordering the endocardium and epicardium contours as well as regions with an area of only 1-pixel were considered as noise and were removed from the ground truth during training and assessment [16].

### Model interpretability

Interpretability is a limitation of most neural networks, such as U-Net due to the large number of convolutional layers used. This effectively renders the neural network a black box, making it difficult to determine the cause of misclassification or inaccurate segmentation. SAUNet addresses the interpretability and robustness constraints by including a secondary shape stream in its architecture in parallel to the regular stream that mainly captures images' texture information. SAUNet includes modules in its architecture that take in intermediate feature maps at multiple stages of the network. These attention maps can be extracted to produce a visual heatmap that can be used for interpretability. More specifically, the heatmap visualization highlights the regions of the input image that have a more substantial impact on the final segmentation and can be used to detect the layers of origin where the inaccuracy of prediction starts for samples with faulty predictions. This opportunity provides the model developers with a better understanding of where to focus on for improving the model.

### Comparison to other models

U-Net architecture variations have been used in other studies to segment CMR LGE images in HCM patients. We analyzed our test dataset using the standard U-Net architecture and compared the results to our model's performance for CMR LGE quantification [13–17]. The 4SD and 6SD U-Net models were implemented in the PyTorch framework and trained with the same hyperparameters and data as the SAUNet models.

### Statistical analysis

Continuous variables were expressed as mean ± standard deviation. Pearson correlation coefficient (r) and Bland-Altman analysis were used to assess agreement between automatic and

manual scar volumes and the %LGE. Overlap between the automatically segmented and the manually segmented areas in samples was measured by using the DSC for both the LV scar and the myocardium. DSC calculates the area of overlap between ground truth and the prediction divided by the total number of pixels in the images:

$$DSC(GT, P) = \frac{2|GT \cap P|}{|GT| + |P|}$$

where $GT$ is the ground truth and $P$ is the model prediction pixels [19]. DSC was only calculated from samples with scar present on either the ground truth or the model prediction as DSC is not defined when there is no scar. DSC scores were classified as follows: less than 0.25, poor; 0.25–0.49, moderate; 0.5–0.74, good; and 0.75 or greater, excellent. We used a nonparametric Kruskal-Wallis test to compare DSC values, the Welch t-test to compare regression slopes, and the Fisher z test to compare correlation coefficients. Statistical significance was defined as $P < 0.05$. Statistical analyses were performed using R 4.0.2 [29–31].

The following comparisons between the ground truth and the model prediction for LV segmentations, LGE mass and LGE% were performed:

1. The SAUNet 4SD model versus the SAUNet 6SD model,

2. Our SAUNet model versus the standard U-Net model using either a 4SD or 6SD cutoff.

Many previous studies on HCM images required image pre-processing with manually cropping to isolate the LV prior to analysis [13]. Thus, we also performed the above comparisons on pre-processed images that were cropped around the LV region to observe the effect of cropping on segmentation. Each 2D image slice was cropped around the epicardium border using the positional information provided in the ground truth epicardium mask. Then, each image was zoomed to a dimension of 160x160 $px^2$ consistent with previous works [17]. Lastly, differences in model performance on our testing set partitioned by sex were examined for the SAUNet 6SD model.

## Supporting information

**S1 Fig. Correlation and Bland-Altman analysis for U-Net 4SD model.** Correlation (a) and Bland-Altman analysis (b) between the expert-based manual analysis and the model prediction for CMR LGE scar mass. Correlation (c) and Bland-Altman analysis (d) between the expert-based manual analysis and model prediction for percentage of LGE volume.
(TIFF)

**S2 Fig. Correlation and Bland-Altman analysis for U-Net 6SD model.** Correlation (a) and Bland-Altman analysis (b) between the expert-based manual analysis and the model prediction for CMR LGE scar mass. Correlation (c) and Bland-Altman analysis (d) between the expert-based manual analysis and model prediction for percentage of LGE volume.
(TIFF)

**S1 Table. Size of training and testing samples grouped by their site and scarring label.**
(XLSX)

**S2 Table. Bland Altman analysis, t-test, and predicted against expert-read mean mass and LGE% for the 6SD SAUNet on the testing set split by sex.**
(XLSX)

**S3 Table. Number of samples grouped by site and software used by experts.**
(XLSX)

## Author Contributions

**Conceptualization:** Zeinab Navidi, Jesse Sun, Raymond H. Chan, Kate Hanneman, Amna Al-Arnawoot, Harry Rakowski, Anna Woo, Bo Wang, Wendy Tsang.

**Data curation:** Raymond H. Chan, Martin S. Maron.

**Formal analysis:** Zeinab Navidi, Jesse Sun, Raymond H. Chan, Amna Al-Arnawoot, Alif Munim.

**Methodology:** Zeinab Navidi, Jesse Sun, Bo Wang.

**Project administration:** Wendy Tsang.

**Resources:** Wendy Tsang.

**Software:** Zeinab Navidi, Jesse Sun.

**Supervision:** Raymond H. Chan, Kate Hanneman, Amna Al-Arnawoot, Harry Rakowski, Martin S. Maron, Anna Woo, Bo Wang, Wendy Tsang.

**Validation:** Zeinab Navidi, Jesse Sun, Wendy Tsang.

**Visualization:** Zeinab Navidi, Jesse Sun, Alif Munim.

**Writing – original draft:** Zeinab Navidi, Jesse Sun, Wendy Tsang.

**Writing – review & editing:** Zeinab Navidi, Jesse Sun, Raymond H. Chan, Kate Hanneman, Amna Al-Arnawoot, Alif Munim, Harry Rakowski, Martin S. Maron, Anna Woo, Bo Wang, Wendy Tsang.

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
