## [Decision Letter · Decision Letter 0]

9 Nov 2022

Interpretable Machine Learning for Automated Left Ventricular Scar Quantification in Hypertrophic Cardiomyopathy Patients.

PDIG-D-22-00269

Dear Dr. Tsang,

We are pleased to inform you that your manuscript 'Interpretable Machine Learning for Automated Left Ventricular Scar Quantification in Hypertrophic Cardiomyopathy Patients.' has been provisionally accepted for publication in PLOS Digital Health.

Best regards,

Rutwik Shah, MD

Guest Editor

PLOS Digital Health

Reviewer Comments (if any, and for reference):

Reviewer's Responses to Questions

**Comments to the Author**

1. Does this manuscript meet PLOS Digital Health’s publication criteria? Is the manuscript technically sound, and do the data support the conclusions? The manuscript must describe methodologically and ethically rigorous research with conclusions that are appropriately drawn based on the data presented.

Reviewer #1: Partly

Reviewer #2: Yes

2. Has the statistical analysis been performed appropriately and rigorously?

Reviewer #1: Yes

Reviewer #2: I don't know

3. Have the authors made all data underlying the findings in their manuscript fully available (please refer to the Data Availability Statement at the start of the manuscript PDF file)?

Reviewer #1: Yes

Reviewer #2: Yes

4. Is the manuscript presented in an intelligible fashion and written in standard English?

Reviewer #1: Yes

Reviewer #2: Yes

5. Review Comments to the Author

Reviewer #1: Due to the relevance of scar load in predicting clinical outcomes, scar quantification on cardiovascular magnetic resonance (CMR) late gadolinium enhancement (LGE) imaging is useful in risk stratifying patients with hypertrophic cardiomyopathy (HCM). The goal was to create a machine learning (ML) model that outlines LV endo- and epicardial boundaries and measures CMR LGE pictures from HCM patients. The study is interesting with strong experimental results.

Reviewer #2: This is a retrospective study that aimed to train a machine learning model to quantify scar o cardiovascular magnetic resonance late gadolinium enhancement (LGE) images in patients with hypertrophic cardiomyopathy. Numerous studies have shown that scar burden predicts clinical outcomes in these patients; currently, scar quantification in patients with HCM is mostly manually performed by radiologists, making it quite subjective, cumbersome, and error prone. Machine learning models that have been developed for automated scar quantification are limited due to the need for image pre-processing as well as their comparison to a single institutional reader as the reference standard. The authors of this study aimed to address these key limitations in this study.

The authors used approximately 2000 images to train the Shape-attentive U-Net (SAUNet) model, and then tested the model on approximately 500 images. They found that the model showed good correlation for both LV segmentation as well as scar quantification, when compared to manual LV segmentation and scar quantification. Furthermore, the model could efficiently perform these tasks using uncropped images and on a rapid timeframe.

There are several notable strengths to this study. First, the images used to train the model were acquired from multiple sites, vendors, readers, and analysis packages making this study unique and more broadly generalizable. Second, images were not pre-processed, making it more relevant to implementation in clinical practice.

There are a few key limitations. First, the database used to train the model was not large in size. The ~2,000 images used to train the model came from only 247 patients, of which only 200 patients had LGE scar. Other studies (for example, Fahmy 2019 Radiology) included images from over 1,000 patients. Second, only two expert readers were used as the reference standard. This study would be strengthened by expanding the pool of expert readers, particularly given the heterogenous and patchy nature of hypertrophic cardiomyopathy scar.

Overall, quantification of scar on CMR LGE images in patients with HCM is a key prognostication tool in assessing risk of sudden cardiac death. Rapid and accurate scar quantification provides important information to cardiologists managing these patients, and this paper offers a novel machine learning model for scar quantification.

6. PLOS authors have the option to publish the peer review history of their article (what does this mean?). If published, this will include your full peer review and any attached files.

**Do you want your identity to be public for this peer review?** For information about this choice, including consent withdrawal, please see our Privacy Policy.

Reviewer #1: No

Reviewer #2: No
